# Modified EfficientNet-B0 Architecture Optimized with Quantum-Behaved Algorithm for Skin Cancer Lesion Assessment

**DOI:** 10.3390/diagnostics15243245

**Published:** 2025-12-18

**Authors:** Abdul Rehman Altaf, Abdullah Altaf, Faizan Ur Rehman

**Affiliations:** 1Johns Hopkins Aramco Healthcare (JHAH), Dhahran 34465, Saudi Arabia; 2Faculty of Computer Science and Information Technology, Universiti Tun Hussein Onn Malaysia, Batu Pahat 86400, Johor, Malaysia; 3Kaafat Business Solution, Riyadh 12331, Saudi Arabia; faizanurrehman@gmail.com

**Keywords:** skin cancer classification, HAM10000, ISIC 2019 and MSLD v2.0 datasets, modified EfficientNet-B0, quantum behaved optimization algorithm, Monte Carlo simulation

## Abstract

**Background/Objectives**: Skin cancer is one of the most common diseases in the world, whose early and accurate detection can have a survival rate more than 90% while the chance of mortality is almost 80% in case of late diagnostics. **Methods**: A modified EfficientNet-B0 is developed based on mobile inverted bottleneck convolution with squeeze and excitation approach. The 3 × 3 convolutional layer is used to capture low-level visual features while the core features are extracted using a sequence of Mobile Inverted Bottleneck Convolution blocks having both 3 × 3 and 5 × 5 kernels. They not only balance fine-grained extraction with broader contextual representation but also increase the network’s learning capacity while maintaining computational cost. The proposed architecture hyperparameters and extracted feature vectors of standard benchmark datasets (HAM10000, ISIC 2019 and MSLD v2.0) of dermoscopic images are optimized with the quantum-behaved particle swarm optimization algorithm (QBPSO). The merit function is formulated by the training loss given in the form of standard classification cross-entropy with label smoothing, mean fitness value (mf^val^), average accuracy (mAcc), mean computational time (mCT) and other standard performance indicators. **Results**: Comprehensive scenario-based simulations were performed using the proposed framework on a publicly available dataset and found an mAcc of 99.62% and 92.5%, mfval of 2.912 × 10^−10^ and 1.7921 × 10^−8^, mCT of 501.431 s and 752.421 s for HAM10000 and ISIC2019 datasets, respectively. The results are compared with state of the art, pre-trained existing models like EfficentNet-B4, RegNetY-320, ResNetXt-101, EfficentNetV2-M, VGG-16, Deep Lab V3 as well as reported techniques based on Mask RCCN, Deep Belief Net, Ensemble CNN, SCDNet and FixMatch-LS techniques having varying accuracies from 85% to 94.8%. The reliability of the proposed architecture and stability of QBPSO is examined through Monte Carlo simulation of 100 independent runs and their statistical soundings. **Conclusions**: The proposed framework reduces diagnostic errors and assists dermatologists in clinical decisions for an improved patient outcomes despite the challenges like data imbalance and interpretability.

## 1. Introduction

This introduction will briefly place the study in a broad context and highlight why it is important. According to the World Health Organization, skin cancer accounts for one in three cancers worldwide with 2–3 million non-melanomas and 132,000 melanomas annually [1]. The most typically diagnosed are basal cell carcinoma, squamous cell carcinoma, and melanoma that kills over 75% of skin cancer patients; however, in the case of early detection the survival rate is more than 90% whereas advanced metastatic stages is 25%. According to the cancer data retrieved, 115,320 instances of melanoma were estimated even in developed countries like United States (US) in 2021 [2], along with an anticipated 11,540 fatalities every year. It is the most frequent and lethal kind of cancer diagnosed globally, with more than 1.5 million new cases predicted in the year 2022; there were more than 0.33 million fresh cases of melanoma identified globally, and almost 25% of people died from the disease [3]. Clinically, skin cancer lesions (SCLs) vary in appearance, size, shape, and color, making manual identification difficult. The traditional methods for SCLs diagnosis mainly include visual inspection, dermoscopy, and histopathological examination, all of which have their own advantages and disadvantages [4]. The traditional methods like dermoscopy and biopsy improve accuracy, but they are subjective, invasive, and time-consuming. These limitations highlight the need for automated and reliable approaches such as computer-aided (CAD) techniques for improved diagnostics.

These CAD techniques gained significant importance due to machine learning (ML) [5] as well as nature-inspired optimization algorithms in the domain of medical diagnostics in general and SCLs in particular. These optimizers involve a stack of algorithms like ant colony optimization (ACO), particle swarm optimization (PSO), genetic algorithms (GA), differential evaluation (DE) and firefly algorithm, which are widely applied for feature selection, parameter tuning, and classification tasks [6,7,8]. ACO mimics the foraging behavior of ants to identify optimal feature subsets, while PSO models the collective movement of birds or fish to achieve faster convergence in optimization. GA employs evolutionary principles such as selection, crossover, and mutation to generate robust solutions, whereas the firefly algorithm uses the attraction behavior of fireflies to enhance exploration and exploitation in complex search spaces. When integrated with ML or deep learning models (DLM) [9,10], these algorithms improve lesion classification accuracy, reduce computational complexity, and enhance early detection, making them promising tools for clinical decision support in dermatology [11,12].

Pretrained DLM have become highly effective in SCLs analysis due to their ability to leverage transfer learning. Instead of training a network from scratch, models such as VGGNet, ResNet, Inception, DenseNet, and EfficientNet are initially trained on large-scale datasets like ImageNet and then fine-tuned on dermoscopic images [13,14,15]. This approach enables the model to reuse learned low-level and mid-level features such as edges, textures, and color variations, which are crucial in distinguishing between malignant and benign skin lesions [16]. By doing so, pretrained models significantly reduce the need for very large annotated medical datasets, which are often difficult and expensive to obtain in dermatology. Moreover, pretrained models have shown state of the art performance in tasks such as lesion classification, segmentation, and feature extraction. For instance, EfficientNet variants and Inception architectures have demonstrated high accuracy on benchmark datasets like HAM10000 [17] and ISIC2019 [18]. These models not only enhance diagnostic reliability but also reduce computational costs and training time, making them more practical for real-world clinical applications. Their success highlights the potential of transfer learning in bridging the gap between limited medical data and the growing need for automated, accurate, and scalable SCLs detection systems. Applying Quantum-behaved PSO (QPSO) on a modified EfficientNet-B0 (MENet-B0) is motivated by the need for higher accuracy and efficiency in skin cancer lesion classification. While EfficientNet-B0 provides a strong baseline, QPSO enhances optimization by effectively exploring high-dimensional search spaces. This integration enables better feature selection, faster convergence, and improved diagnostic reliability, making it highly suitable for clinical applications. SCLs datasets present several challenges that make automated analysis and classification difficult. One of the major issues is class imbalance, as benign lesions are far more common than malignant ones, leading to biased learning in models [19]. Additionally, the availability of high-quality annotated data is limited, since expert labeling and biopsy confirmation are required. Another difficulty lies in the high intra-class variability, where lesions of the same type may appear very different, and low inter-class variability, where different lesion types often look visually similar. Image quality issues such as variations in lighting, resolution, and the presence of artifacts like hair or ruler marks further complicate analysis. Moreover, the lack of diversity in terms of skin tones, demographics, and geographical representation limits the generalizability of models [20]. Together with privacy and ethical constraints on medical data sharing, these challenges highlight the complexity of building robust and reliable skin cancer lesion classification systems. The main contribution of the undertaken research is as follows:Imbalanced datasets can bias the model towards the higher sample class that is addressed by synthetic minority oversampling technique.The resemblance of numerous skin lesions is controlled by introducing the mobile inverted bottleneck convolution with a squeeze and excitation approach in efficentNet-B0 [21] framework, as correctly classifying lesions can be problematic due to their similarity.QBPSO optimizer is exploited to reduce the redundant features by managing the overfitting and optimizing the hyperparameters of MENet-B0 architecture to increase the framework’s precision, accuracy and produce a lightweight architecture.The reliability of the proposed framework is guaranteed based on various performance indicators like mf^val, mAcc and mCT by applying Monte Carlo simulations based on 100 independent executables.The results of the proposed framework are compared with state-of-the-art pre-trained models based on EfficentNet-B4 [22], RegNetY-320 [23], ResNetXt-101 [24], EfficentNetV2-M [25], VGG-16 [26], Deep Lab V3 [27] as well as reported techniques based on Mask RCCN [28], Deep Belief Net [29], Ensemble CNN [30], SCDNet [31] and FixMatch-LS [32] techniques on HAM10000 and ISIC2019 datasets.The stability of the proposed optimizer is established based on the merit function fitness profile and various ablations studies on feature selection mechanism.

The rest of this article is organized as follows. Literature survey and state-of-the-art techniques on SCLs are presented in Section 2. The details on datasets and their associated difficulties along with MENet-B0 architecture and QBPSO mathematical explanation is elaborated in Section 3. Section 4 depicts the comprehensive simulations based on various ablation studies, comparison with pre-trained models and state-of-the-art existing techniques along with parameter values and settings used for the simulations. In the last section, conclusions are drawn based on the scenario-based results obtained in the form of various graphs and tables. The limitations of the existing framework and directions for future work is also provided for future researchers.

## 2. Literature Survey and State-of-the-Art

SCLs analysis is one of the most sensitive techniques for researchers of existing cancers due to its diagnostic accuracy. Clinical visual inspection using the ABCD rule or the 7-point checklist provides accuracy ranging from 65% to 80% [33], depending on clinician experience, while dermoscopy-based examination can raise diagnostic accuracy up to 85% in expert hands. Early CAD systems based on handcrafted features and classical classifiers reported classification accuracy in the range of 70–85% on benchmark datasets such as PH2 and early ISIC releases [34]. The methods using color histograms, border irregularity, and texture descriptors with support vector machines (SVMs) achieved around 78–82% sensitivity and 80–85% specificity in distinguishing melanoma from benign nevi [29]. Segmentation methods like thresholding and active contours achieved Dice scores between 0.75 and 0.85, indicating reasonable but imperfect lesion boundary delineation [35]. These quantitative results demonstrate that while traditional approaches provide interpretable and computationally inexpensive solutions, they often struggle to reach the high reliability needed for clinical adoption.

Despite these limitations, the value of traditional techniques lies in their interpretability and role in setting benchmarks for newer models. For instance, Gray Level Co-occurrence Matrix (GLCM) and local binary pattern (LBP)-based texture analysis combined with random forests achieved approximately 80% balanced accuracy, but performance degraded by 10–15% when tested on cross-dataset scenarios, highlighting the generalization issue [36]. Similarly, handcrafted features are highly sensitive to variations in lighting and acquisition, often reducing classification accuracy by 5–10% when evaluated across heterogeneous datasets. These shortcomings explain why histopathology, with diagnostic accuracy exceeding 95%, remains the gold standard for confirming melanoma and other malignancies [37]. Nevertheless, traditional approaches have provided a critical foundation for the development of automated systems, offering both interpretable features and performance benchmarks that guided the transition to deep learning models surpassing 90–95% accuracy in recent years [38]. Much work has been performed by researchers to classify SCLs with different ML and DL techniques; some of the notable contributions are provided here as references. To this end, Filali et al. developed an interest in and made efforts to create automatic diagnostic procedures that might aid in early detection. Several approaches based on ML and DL have been proposed to help in the diagnosis of skin lesions. The main purpose of this research is to undertake a comparative evaluation of several DL architectures involving ResNet, Google Net, VGG16Net, and AlexNet for classifying skin cancer into malignant and benign. In the results they found from their in-comparison research, the ResNet architecture outperformed the others in terms of classification accuracy, and this is extremely encouraging [39]. Srinivasa Rao et al. [40] proposed a Generative Adversarial Network (GAN) segmented model that input images into different regions or classes corresponding to different skin lesion types; this provides insights into how the GAN model separates different regions of interest in the image, also proposed across multiple epochs. The GAN-generated images are also passed through the trained Deep Learning Convolutional Neural Network (DLCNN) model for prediction, providing valuable information for early diagnosis and treatment planning. Naeem et al. [41] performed integration of handcrafted techniques with various models of DL like EfficientNetB0, B1 [42], MobileNetV2 [43], DenseNet-121 and ResNet101 [44] to compare their effectiveness. These models outperform other state-of-the-art techniques like the Grad-CAM heat map method, random forest algorithm and standard convolutional neural networks (CNN) for categorizing multiple classes of SCLs. Khan et al. [28] evaluated the DL model based on Mask RCNN on three publicly available standard datasets—ISBI2016, ISBI2017, and HAM10000—and presented the results of various performance indicators like sensitivity, precision, accuracy and F1-Score for both segmentation and classification.

Moreover, an optimized approach is described for diagnosing SCLs from standard datasets by applying a CNN model whose adaptive variables are optimized with a whale optimization algorithm to minimize the error and obtain the desired output. Kaseem et al. [45] used transfer learning on a pre-trained DL network based on modified Google Net by adding more filters and transforming the model’s layers to perform binary and muti-class classifications on an ISIC2019 dataset. Similarly, Pandiyan et al. [46] detected skin lesions using discrete wavelet transformation (DWT)-based feature extraction techniques in which optimal features of DWT are fed into CNN with SoftMax activation function. The effectiveness of the suggested model was tested through the HAM10000 dataset, which produced remarkable accuracy compared to the existing ML models. Zafar et al. [23] developed CAD systems to assist dermatologists by providing accurate and reliable analysis of SCLs based on DL techniques for classification. The system was implemented on a publicly accessible (HAM10000) database and performed an in-depth analysis using standard performance indicators on benchmarked DL models. Mushtaq et al. [47] defined a framework of comprehensive categorization for the multi-class classification of skin cancer carcinoma by employing CNN hybridized with ensemble visual geometry group-16 (EVGG-16). The average accuracy of 87% is found with an F1-score of 0.89 on HAM10000 dataset. Prasad et al. [48] refined the application of the DL technique to achieve excellence in the detection of skin cancers that exploit the EfficientNet-B3 method; however, they have pre-processed the HAM10000 dataset by performing contrast enhancement, rescaling, and an arbitrary ±20% brightness modification to obtain an accuracy of 90.6%. The baseline contribution by various researchers is listed in Table 1 along with the details of the DL models used for multiclass SCLs.

Moreover, a few of the latest contributions on Mpox Skin Lesion Dataset Version 2.0 (MSLD v2.0) dataset are also taken into account. In this regard, various models of ConvexNeXt techniques have been applied to this recently developed multiclass dataset and an overall accuracy of 94.0% was observed [49]. Similarly, several transformer-based architectures have demonstrated strong performance in SCL classification tasks. Elhadidy et al. [50] (2025) evaluated multiple transformer variants and reported promising results; where the standard Vision Transformer achieved an accuracy of 93.10%, the DINO self-supervised model attained 90.40%, and the Swin Transformer outperformed the others with an accuracy of 93.71%. Similarly, Alghoraibi et al. [51] (2025) explored hybrid transformer–CNN architectures and observed that the ResNetViT model achieved an accuracy of 92.00%, while their ViT hybrid model slightly improved performance to 92.19%.

Although much work has been performed in this domain, there is still room for improvement in terms of methodology, computational cost and accuracy.

**Table 1 diagnostics-15-03245-t001:** Baseline reported and pre-trained models used for multiclass SCLs characterization.

Reference	Methodology	Datasets	Accuracy (%)
Khan et al. [28], 2021	Mask RCCN	HAM10000	88.5
Ali et al. [22], 2022	EfficentNet-B4		87.9
Zafar et al. [23], 2022	RegNetY-320		92.7
Chanrvedi et al. [24], 2023	ResNetXt-101		93.0
Venugopal, V. [25], 2023	EfficentNetV2-M		94.8
Sukanya et al. [29], 2023	Deep belief Net		94.0
Tahir et al. [52], 2023	DSCC-Net		94.17
Pacheco et al. [26,30], 2019, 2020	Ensemble CNN VGG-16	ISIC2019	90.1 82.5
Naeem et al. [31], 2022	SCDNet		92.91
Aldhyani et al. [27], 2022	Deep LabV3		89.5
Gilani et al. [53], 2023	Spiking VGG-13		91.81
Zhou et al. [32], 2023	FixMatch-LS		94.0
Waqar, M. et al. [49], 2025	ConvexNeXt Models	MSLD v2.0	94.00
Elhadidy, M.S. et al. [50], 2025	Vision transformer		93.10
	Dino		90.40
	Swin Transformer		93.71
Alghoraibi, H. et al. [51], 2025	ResNetViT		92.00
	ViT Hybrid Model		92.19

## 3. Materials and Methods

This section consists of four main segments; in the first part the detail of the datasets is presented along with their associated difficulties. In the second segment, adaptive DL architecture based on modified efficient Net B0 is formulated along with the fitness evaluation function formulation in terms of supervised manner. It involves adapting neural networks trained on massive databases, like HAM10000 and ISIC2019 for computer vision, to perform tasks when labeled data or computational resources are restricted. In the last part, a quantum-behaved PSO learning protocol is provided along with different performance measures like fitness value, computational cost, accuracy and reliability. The overall framework used for muti-class classification of the skin cancer lesion is given below in Figure 1.

### 3.1. Diagnostic Categories and Image Details of Skin Lesion Datasets

Three complex popular datasets like HAM10000 [28], ISIC 2019 [31] and MSLD v2.0 [50] of skin lesions are exploited to examine the strength of the proposed architecture. The HAM10000 dataset is used for medical image analysis of skin cancer with 10,015 high-quality dermatoscopic images available publicly, representing seven different classes of pigmented skin lesions. These includes actinic keratoses and intraepithelial carcinoma (AKIC) with 327 images, which are precancerous cancerous lesions, and basal cell carcinoma (BCC) with 514 images, a common form of skin cancer.

The dataset also contains 1099 images of benign keratosis-like lesions (BKL) such as seborrheic keratoses, and dermatofibroma (DF) having 115 images, which are harmless fibrous nodules. Another critical class with 1113 images is melanoma (MEL), the most dangerous type of skin cancer due to its high potential to spread. A few of the images from the HAM10000 dataset are given in Figure 2. The second dataset used in this study is a prominent and publicly recognized dataset called ISIC 2019, a few samples from which are given in Figure 3, that is used as a benchmark to evaluate how well the proposed models can distinguish between several skin lesions. It has eight distinct classes like melanoma (Mel), melanocytic nevus (MelN), basal cell carcinoma (BaCC), actinic keratosis (ActK), benign keratosis (BeKer), dermatofibroma (DerF), vascular lesion (VasL) and Squamous cell carcinoma (SCC), respectively, having a total of 25,331 dermoscopic images. The datasets are naturally imbalanced, as in HAM10000 the disease MV constitutes 67 images of the dataset while DF and VL have 115 and 142 images, respectively. Similarly, ISIC 2019 suffers an even higher imbalance than that of HAM10000, which reflects real-world clinical data distribution but poses a major challenge for training balanced and accurate AI models. The MSLD v2.0 dataset is also taken in this study, which is the latest dataset and quite promising with respect to its attributes. It has six different classes like monkeypox (Mpox); chickenpox (chipox); measles (Mea); cowpox; hand, foot, and mouth disease (HFMD); and healthy; a few samples from which are given in Figure 4.

To handle the imbalanced dataset issue, synthetic data generation techniques based on Synthetic Minority Over-sampling Technique (SMOTE) [54] are applied to create samples of the rare classes. A total of 2000 images per class have been formulated to avoid the size of synthetic sets and risk of overfitting, except for the MV class.

Therefore, 8690 synthetic images are generated by the SMOTE, leading the HAM10000 dataset to a total number of 18,705 images. Similarly in the dataset of ISIC2019, the class MN is dominant with 12,875 images, while the images of DerF and VasL have 239 and 253 images, respectively, which make the datasets notoriously imbalanced. A perfect sample size of 4000 images per class was formulated by the SMOTE, other than the MN class, to maintain a realistic scenario. The image size and bit depth of both the datasets is uniformly 224 × 224 and 24, respectively.

Table 2 presents the dataset distribution and characteristics of the HAM10000, ISIC2019 SCLs and MSLD v2.0, in which the datasets are divided into training (70%) and testing (30%), split across multiple lesion classes. HAM10000 contains seven lesion categories with varying sample sizes, where the largest class is MV with 6705 images. ISIC2019 includes eight classes, with MN being the largest, comprising 12,875 samples. Moreover, the distinct classes of MSLD v2.0 are six, including the healthy features. All images are standardized to a resolution of 224 × 224 with a 24-bit depth, ensuring uniformity in input size for model training and evaluation. This structured distribution highlights dataset diversity and balance, crucial for robust classification of SCLs.

### 3.2. Designed MENet-B0 Architecture and Fitness Function Formulation

The proposed MENet-B0 architecture begins with an input image that passes through an initial 3 × 3 convolutional layer to capture low-level visual features such as edges and textures. The core feature extraction is performed using a sequence of Mobile Inverted Bottleneck Convolution (MBConv) blocks, where both 3 × 3 and 5 × 5 kernels are employed to balance fine-grained detail extraction with broader contextual representation. In this design, multiple MBConv6 layers are stacked sequentially to increase the network’s learning capacity while maintaining computational efficiency. To further optimize representation, a 1 × 1 bottleneck convolution reduces dimensionality before passing features to the classification head. Global Average Pooling (GAP) and a flattening operation are applied to compress the spatial information into a compact feature vector. A key modification in this architecture is the integration of a self-attention mechanism, which enables the network to focus selectively on discriminative lesion regions and suppress irrelevant background information. The extracted features are then passed to a fully connected dense layer, followed by a Softmax activation function for multi-class probability distribution. A novel EfficientNet-B0 (NEN-B0) is developed based on mobile inverted bottleneck convolution (MB Conv) with a squeeze and excitation (SqEx) approach. Consider X is an input feature map extracted from $X\in R^{H\times W\times C^{inp}}$. The expansion of 1 × 1 convolution is performed by using the expression given in Equation (1).(1)Z=gX∗Wexp,Wexp∈R1×1×Cinp.×tCinp,gt=t1+exp(−t)
where t is the expansion ratio and g(.) is the swish activation function. The depthwise convolution is performed using a specified size of the kernel using Equation (2) and having the same activation function g(.), which is a smooth nonlinear function that improves gradient flow, polynomial approximations and learning accuracy.(2)$$Z_{dw}=g\left(Z⨀W_{dw}\right),W_{dw}\in R^{k\times k\times tC^{inp}}$$
where ⨀ denotes depthwise convolution applied channel-wise with a kernel size of k. The three operations have been performed during the process of SeEx that are global average pooling “℘”, channel attention “CA” and recalibration “$Z_{SqEx}$” as calculated in Equations (3)–(5), respectively.(3)$${℘}_{c}=\frac{1}{HW}\sum_{i=1}^{H}\sum_{j=1}^{W}Z_{dw}\left(i,j,c\right)$$(4)CA=oW2ℏ(W1℘), o.=ReLUℏ(.)=11+exp(−t)(5)$$Z_{SqEx}\left(i,j,c\right)={CA}_{c}.Z_{dw}(i,j,c)$$

Now the projection on the matrix W with 1 × 1 convolution is performed on $Z_{SqEx}$ by using the relation provided in Equation (6).(6)$$Z_{SqEx}*W_{proj},\;\;W_{proj}\in R^{1\times 1\times tC^{inp}\times C^{outp}}$$

The residual connection is established by having the assumptions that $C^{inp}=C^{outp}$ and value of the stride = 1. Moreover, the compound scaling factors are also introduced below to achieve a balanced depth (dept), width (wid) and resolution (reso).(7)$$\left\{\begin{array}{c} dept=\alpha^{\vartheta }\\ wid=\beta^{\vartheta }\\ reso=\gamma^{\vartheta } \end{array}\right.,\;\;\alpha .\beta^{2}.\gamma^{2}\approx 2$$
where ϑ is the user-defined scaling coefficient that controls the model size and α, β, γ are adaptive constants. This ensures that Floating Point Operations (FLOPs) roughly double for each increment of ϑ.

The merit function is formulated by the training loss “${fit}_{L}$” given in the form of standard classification cross-entropy with label smoothing given in Equation (8).(8)$${fit}_{L}=-\sum_{i=1}^{N}\sum_{c=1}^{C}y_{ic}\log\left({\hat{y}}_{ic}\right)$$
where ${\hat{y}}_{ic}=\frac{exp(Z_{ic})}{\sum_{j=1}^{c}exp(Z_{ij})}$ and $y_{ic}$ are smoothed one-hot labels. Overall, the architecture extends the original EfficientNet by employing mobile inverted bottleneck convolution (MB Conv) blocks, which first expand the input channels, apply computationally efficient depthwise convolutions, and then project them back to lower dimensions. Each block is further enhanced with a Squeeze-and-Excitation (SE) mechanism that adaptively recalibrates channel-wise features by emphasizing informative channels and suppressing less relevant ones. Instead of the standard ReLU, Efficient Net uses Swish, which is smoother and helps the model learn better patterns. Unlike traditional models that scale depth, width, or resolution independently, EfficientNet introduces a compound scaling method that uniformly balances all three dimensions, enabling higher accuracy without unnecessary computational overhead. Moreover, the proposed architecture introduces deeper MBConv stacking and attention-based refinement, thereby improving its ability to handle the complex textures, shapes, and color variations typical in SCLs images. After the average pooling layer and before the flatten layer, adding a self-attention layer is intended to improve the model’s fitness to capture multifaceted patterns in the data. The fully connected (FC) layer is responsible for learning complex patterns by adjusting weights through training. Furthermore, the hyperparameters of MENet-B0 are optimized with a QBPSO algorithm to reduce the feature vector size as well achieving the optimal setting for the architecture to have minimum learnable parameters and computational time without compromising the level of accuracy. The architecture of proposed MENet-B0 is given below in Figure 5 while the mathematical details regarding QBPSO are presented in the next section.

### 3.3. Quantum Behaved Particle Swarm Optimization Algorithm

The standard particle swarm optimization (PSO) algorithm was inspired by birds’ social behavior, in which the learning is performed within the generation. Kennedy and Eberhart [55] were the scientists who presented this idea in 1995, and since then many versions have been presented by the researchers, applying it in different domains in science and engineering [56,57,58]. Keeping in view the simplicity of coding and strength of this population-based algorithm, a new version based on quantum mechanics called quantum-behaved particle swarm optimization (QBPSO) is exploited to not only optimize the hyperparameters but also for tuning feature vectors of SCLs datasets. QBPSO transforms the velocity and position updating mechanism of a standard PSO into a quantum-inspired probabilistic scheme. In the algorithm each particle is treated as a quantum particle confined in a potential well; its position is sampled from an analytical probability distribution derived from the quantum model. This yields a simple position update with only one control parameter (contraction-expansion coefficient) and strong global search behavior rather than having velocity and position equations separately.(9)$$p_{j}\left(t\right)=\xi ⨂{Lbest}_{j}+(1-\xi )⨂\;Gbest,\;\;\;\;\;\;\xi ∼U(0,1)$$
where ⨂ denotes elementwise multiplication.(10)$$mbest\left(t\right)=\frac{1}{M}\sum_{j=1}^{M}{Lbest}_{j}$$(11)$$p_{j,d}\left(t+1\right)=p_{j,d}\left(t\right)\pm \psi \left|mbest\left(t\right)-p_{j,d}\left(t\right)\right|.ln\left(\frac{1}{\mu_{j,d}}\right)$$(12)$$p_{j,d}\left(t+1\right)=p_{j,d}\left(t\right)+s.\psi \left|mbest\left(t\right)-p_{j,d}\left(t\right)\right|.ln\left(\frac{1}{\mu_{j,d}}\right)$$

The standard performance measures well-known in the literature [59,60,61] are used to see the effectiveness of the proposed architecture; these measures involve precision, recall, False Negative Rate (FNR), F1-score, specificity and False Positive Rate (FPR). Moreover, the reliability of the QBPSO and complete framework is exploited using statistical performance indicators like minimum of 100 independent runs (MIN), maximum of independent runs (MAX), mean (μ), standard deviation (Std) and kurtosis (Kur) and Matthews Correlation Coefficient (MaCC) [62,63]. The mathematical formulas for μ and Std are used as standard while those for Kur and Mcc are provided in Equations 13 and (14), respectively.(13)$$Kur=\frac{IR(IR+1)}{\left(IR-1)(IR-2)(IR-3\right)}\sum_{i=1}^{100}{\left(\frac{x_{i}-\mu }{Std}\right)}^{4}-\frac{3{\left(IR-1\right)}^{2}}{\left(IR-2)(IR-3\right)}$$(14)$$MaCC=\frac{\left(TP\times TN\right)-\left(FP\times FN\right)}{\sqrt{\left(TP+FP\right)\left(TP+FN\right)\left(TN+FP\right)\left(TN+FN\right)}}$$
where $x_{i}$ is the result at the *i*th run, and IR is an independent run that has been taken as 100 for a comprehensive analysis of the stochastic-based architecture. The results of these 100 executables are stored in the database to perform the statistical analysis. These statistical analyses play a vital role in stochastic-based algorithms because their performance inherently depends on randomness and probability rather than deterministic rules. Since such algorithms may yield different results in different runs, statistical measures such as mean, variance, standard deviation, and confidence intervals are essential to evaluate their overall performance and robustness. Ultimately, statistical analysis provides reliability and interpretability, transforming the uncertainty of stochastic outcomes into meaningful insights, which is crucial for validating these algorithms in practical and real-world applications.

The learning processes used in the study are given in the form of pseudocode.


Start
          X-raw, 
Y-raw = loaddataset ([HAM10000], [ISIC2019])     // load dataset and 
pre-processing 
          X-pre, Y 
= PreprocessImages(X-raw, Y-raw) 
          X-aug, 
Y-aug = AugmentTrainingImages(X-pre, Y)     // Using SMOTE 
          FeEx = 
MENet-B0featureExtractor()      // Feature extraction using MENet-B0 
architecture 
          [n-sample, 
n-features] = FeEx.extract(X-aug)  
          Formulate 
the fitness function 〖fit〗 _L using Equation (8)
          [best-solution, 
fval, time] = QBPSO-FeatureSelection(〖fit〗 _L, Y-aug, 
n-Features, M = 100, MaxIter = 350, λ= 0.01)      // Applying QBPSO 
via probability mapping for feature selection
          QPSO-FeatureSelection(n-sample, 
n-features, Y, D, N, MaxIter, λ)
          for j = 1: 
M

x[j] = RandomUniform(−1, 1, size = D)       

pbest[j] = x[j]

pbest-score[j] = −inf

mbest = Mean(pbest[1:M], axis = 0)
          end
          iter = 0
          while 
iter < MaxIter

beta = 0.9 x (1 − iter/MaxIter) + 0.1     // decreases from ~1 to 0.1
          for j = 1: 
M

S[j] = ContinuousToBinary(x[j])  

score = EvaluateFitnessSelection(n-sample, n-features, Y, S[j], λ)
          end
          if (score 
> pbest_score[j])

pbest[j] = x[j].copy()

pbest-score[j] = score

gbest-jdx = argmax(pbest-score)

gbest = pbest[gbest-jdx]

gbest-score = pbest-score[gbest-jdx]

mbest = Mean(pbest[1, …, M], axis = 0
          else

Continue;  
          u = 
RandomUniform(0, 1, size = D)
          phi = 
RandomUniform(0, 1, size = D)
          P[j] = 
phi * pbest[j] + (1 − phi) * gbest
          for d = 1 
to D
          u-d = 
u[d]
          if 
(RandomUniform(0, 1) < 0.5)

x[j][d] = P-i[d] + beta * abs(mbest[d] − x[j][d]) * log(1/u − d)
          else

x[j][d] = P-i[d] − beta * abs(mbest[d] − x[j][d]) * log(1/u − d)
          end 
          final-score, 
test-metrics = FinalTrainTestEvaluation(F[:, selected-jdx], Y-aug, DATASET)
          return 
selected-idx, final-score, test-metrics
          End


## 4. Simulation and Results

The proposed framework is validated on two publicly available datasets, HAM10000 and ISIC19 [17,18], whose images depict a replica of real-time skin lesion snapshots. Scenario-based comprehensive simulations are performed to analyze the dynamics of skin cancer using a sufficient number of graphical and numerical illustrations supported with ablation studies and comparative analysis with state-of-the-art reported results. The software environment is MATLAB^®^ version R2023b, while the hardware specifications are an Intel (R) Core processor i9, 64 GB of RAM with an 8 GB NVIDIA RTX graphics card. The dataset is split randomly into a ratio of 70:30 in such a way that 70% of the images of each class are used as training while 30% are saved for framework testing. An input of 384 × 384 × 3 is passed to the proposed architecture with a patch size of 16 × 16, dropout rate of 0.2 and an embedding dimension set to be 1024 with 24 layers. The hyperparameter values of MENet-B0 optimized with QBPSO are tabulated in Table 3 while the QBPSO adaptive parameters are provided in Table 4.

A scenario-based evaluation for all three datasets is performed to see the strength of the proposed framework and the results are presented in the form of confusion matrices and tables. In scenario-I, a HAM10000 dataset with seven interdependent classes are examined and its confusion matrix is presented in Figure 6 to summarize the classification performance across seven SCLs categories: AKIC, BCC, BKL, DF, MEL, MV, and VL. Each row corresponds to the true class while each column represents the predicted class, with the diagonal cells showing correct predictions and off-diagonal cells indicating misclassifications. The model demonstrates excellent performance, achieving an overall accuracy of 99.62% with only 0.38% error. Per-class accuracy is consistently high, ranging between 99.45% and 99.81%. For instance, AKIC was correctly classified in 1989 out of 2000 cases (99.45%), BCC in 1991 cases (99.55%), and MEL in 1992 cases (99.60%), while the largest class, MV, achieved 6692 correct predictions out of 6705 (99.81%). Misclassifications are minimal, distributed sparsely across other categories without any strong bias toward a particular incorrect class. This indicates that the classifier is not only highly accurate but also robust and balanced across different lesion types, ensuring reliable diagnostic performance.

Moreover, the classification results based on various performance measures like precision, recall, FNR, F1 score, specificity and FPR for the HAM10000 dataset are tabulated in Table 5. From the table, it is quite evident that precision and recall values remain above 0.99 for every class, which indicates that the model is highly effective in correctly identifying positive cases while minimizing false detections. The F1-scores, which provide a balanced measure of precision and recall, also remain very high, ranging between 0.9938 and 0.9984, confirming the overall reliability of the predictions. The FNR is extremely low, with values between 0.0019 and 0.0055, suggesting that very few actual cases are missed by the classifier. Similarly, the FPR is negligible, staying well below 0.001 for all classes, while the specificity values are close to perfect (≥0.999), reflecting the model’s strong ability to correctly reject negative cases. Notably, the MV class achieves the best balance with the highest F1 score (0.9984) and the lowest FNR (0.0019). Overall, the table demonstrates that the classifier achieves outstanding, balanced, and reliable performance across all lesion categories, making it highly effective for accurate skin lesion classification.

Overall accuracy is found to be 99.62% with a misclassification rate of 0.38% and weighted F1 score is found to be 0.9962.

In scenario II, the ISIC2019 dataset having resolution 256 × 256 is given as an input to the proposed framework by using the parameter values and settings presented in Table 2 and Table 3. The confusion matrix for the ISIC2019 dataset is given below in Figure 7 that illustrates the classification performance across eight classes: ActK, BaCC, BeKev, DerF, Mel, MN, SCC, and VasL, based on a dataset of 40,875 samples.

The diagonal cells represent correctly classified cases, while the off-diagonal entries highlight misclassifications. Overall, the model achieved an accuracy of 92.50%, correctly classifying 37,811 instances while misclassifying 3064. Class-wise results show variability: BaCC attained the highest accuracy of 95.53%, followed by VasL at 95.67% and DerF at 94.95%, reflecting strong performance in these categories. Mel and ActK also performed well, with accuracies of 93.38% and 92.17%, respectively. In contrast, BeKev and SCC exhibited lower accuracies of 89.98% and 91.80%, indicating higher misclassification rates, while MN showed the weakest performance at 90.66%, largely due to a higher confusion with other classes. These outcomes suggest that although the classifier performs robustly across most categories, certain classes such as MN and BeKev remain more challenging, requiring further refinement to reduce misclassification and enhance balanced performance. The level of the precision and recall of the proposed framework is presented in Table 6, which summarizes the performance measures of the classifier across eight lesion categories, showing both strengths and weaknesses in predictive ability. Precision values range from 0.8960 to 0.9864, indicating that most predicted positives are correct, with BeKev and MN achieving the highest precision. Recall values vary between 0.8998 and 0.9567, reflecting the model’s sensitivity, with VasL and BaCC performing best, while BeKev and MN show comparatively lower recall. The FNR is lowest for VasL (0.0433) and BaCC (0.0447), suggesting few missed detections in these classes, but higher for BeKev (0.1002) and MN (0.0934), where the model struggles to capture all true positives.

F1-scores, which balance precision and recall, are consistently strong across classes, ranging from 0.8881 for BeKev to 0.9417 for MN. Specificity values are close to 0.99 across all categories, showing the model’s strong ability to correctly identify negative cases, while the FPR remains low, under 0.012 for every class. Overall, the classifier demonstrates reliable and balanced performance across the different lesion categories, though BeKev and MN exhibit relatively weaker recall, highlighting areas for potential improvement. Overall accuracy is found to be 92.50% with a misclassification rate of 7.50% and weighted F1-score is found to be 0.9253.

In scenario III, MSLD v2.0 dataset with six distinct classes is analyzed by the proposed framework using the same parameter values and settings tabulated in Table 3 and Table 4; the results are presented in Figure 8. The confusion matrix for MSLD dataset is given in Figure 8, which illustrates the classification performance of six classes—Mpox, Chipox, Mea, Cowpox, HFMD and Healthy—while the level of precision and other performance measures is shown in Table 7.

The overall accuracy is found to be 98.30 with a misclassification rate of 0.0170. Table 7 summarizes the classification performance across six classes, showing consistently strong results for all categories. Mpox, HFMD, and Healthy classes achieved near-perfect precision, recall, and F1 scores, indicating highly reliable detection with minimal false errors. Chipox, Mea, and Cowpox also demonstrate high accuracy, with precision and recall values above 0.93, reflecting robust model performance even for more challenging classes. Very low FNR and FPR, along with high specificity, further confirm the model’s strong ability to correctly identify each class while avoiding misclassification. Moreover, it is also worth mentioning that the recall values vary between 0.8998 and 0.9567, reflecting the model’s sensitivity.

### 4.1. Ablation Studies and Tolerance in Proposed Model

The tolerance of the proposed framework is examined by performing various ablation studies by varying one attribute and fixing the rest of them.

#### 4.1.1. Ablation Study 1: Impact of Learning Rates on Various Performance Measures

Table 8 presents the performance of different learning rate strategies, both fixed and adaptive, on the HAM10000 and ISIC2019 datasets, highlighting how optimizer choice impacts classification outcomes. For the HAM10000 dataset, using a fixed learning rate of 0.01 yields an accuracy of 0.8812 with high recall (0.9735) and F1 score (0.9745), while reducing the rate to 0.001 improves accuracy (0.9018) but slightly reduces recall and F1 score. Among adaptive optimizers, AdaGrad achieves balanced performance with an accuracy of 0.9012 and precision of 0.9254, while Adam and Nadam further enhance precision (0.9734 and 0.9032, respectively) and maintain strong recall values above 0.93. RMSProp performs best overall with the highest accuracy of 0.9345, along with precision (0.9453), recall (0.9684), and F1 score (0.9620), showing its robustness for this dataset.

For the ISIC2019 dataset, a fixed learning rate of 0.01 gives lower accuracy (0.8235) but high recall (0.9274), while reducing the rate to 0.001 improves accuracy to 0.8534 with a balanced F1 score of 0.9024. Adaptive methods outperform fixed rates, with AdaGrad achieving accuracy of 0.8835 and high precision (0.9353). Adam and Nadam both show strong recall values above 0.94, though Adam achieves slightly better precision (0.9346). RMSProp again stands out, reaching the highest accuracy (0.9023), along with strong precision (0.9219), recall (0.9550), and the best F1 score (0.9750). Overall, the results indicate that adaptive learning rates, especially RMSProp, consistently provide superior performance across both datasets compared to fixed learning rates. The results are also presented in the form of 3D graphs for the HAM10000 dataset in Figure 9a and ISIC2019 dataset in Figure 9b to show the clear representation of the results.

The 3D bar graph of Figure 9 illustrates the impact of different learning rate variations and adaptive optimizers on classification performance across multiple performance indicators. The x-axis represents the performance indicators (accuracy, precision, recall, and F1-score), while the y-axis denotes the learning rate variations, including fixed rates (α = 0.01, α = 0.001) and adaptive methods (AdaGrad, Adam, NAdam, RMSProp). The z-axis corresponds to the performance level. From the visualization, it is clear that fixed learning rates result in relatively lower performance across most indicators compared to adaptive learning strategies. Among the adaptive methods, RMSProp (yellow) consistently achieves the highest performance across all indicators, followed closely by Nadam and Adam, which maintain strong and balanced results. AdaGrad also shows improvement over fixed learning rates but performs slightly lower than Adam and Nadam. Overall, the figure emphasizes that adaptive optimizers, especially RMSProp, significantly enhance model performance compared to fixed learning rates across all major evaluation metrics.

#### 4.1.2. Ablation Study 2: Impact on Various Optimizers on the Fitness Value of the Deep Learning Model

Keeping in view the stochastic nature of the DL and population-based algorithms, one good result is not enough to support the results; in this regard the fval of the optimizers like SDGM, S-PSO and QBPSO is performed on 100 independent runs, and the results are shown in Figure 10. The results are graphically presented, in which *f*^val^ of the optimizers is drawn on semi-log scale to show the clear difference in each of the executable runs. It is quite clear from Figure 10a that the *f*^val^ achieved on HAM10000 lie in the range from 10^−2^ to 10^−4^, 10^−5^ to 10^−6^ and 10^−9^ to 10^−10^ for SDGM, S-PSO and QBPSO, respectively. Similarly, for the ISIC2019 dataset that is relatively challenging, the *f*^val^ is observed to be in the range 10^−1^ to 10^−3^, 10^−4^ to 10^−6^ and 10^−7^ to 10^−9^ for SDGM, S-PSO and QBPSO, respectively, as shown in Figure 10b.

#### 4.1.3. Ablation Study 3: Comparison of Various EfficientNet B0 Models with the Proposed MENet-B0

Table 9 presents a comparative evaluation of different EfficientNet models and the modified EfficientNet-B0 (MENet-B0) across the HAM10000 and ISIC2019 datasets.

On the HAM10000 dataset, all EfficientNet variants achieved consistently high accuracy, ranging from 97.34% to 98.12%, with EfficientNet-B7 performing best among the baseline models. However, MENet-B0 outperformed all versions, reaching an exceptional accuracy of 99.62% with the lowest F value, reflecting its strong statistical significance. On the ISIC2019 dataset, accuracies were slightly lower across all models, with values ranging from 82.35% for EfficientNet-B0 to 88.99% for EfficientNet-B7. Again, MENet-B0 surpassed all others, achieving 92.50% accuracy with superior F values, confirming its robustness and reliability across different datasets. Overall, the results demonstrate that while deeper EfficientNet models improve performance, the modified MENet-B0 offers the most significant gains in both datasets.

#### 4.1.4. Ablation Study 4: Computational Complexity in Terms of Time in Seconds for Different Optimizers

Computational time is crucial in DL model optimization through heuristic or population-based techniques because it reflects the efficiency and practicality of the approach because this hybrid is only fruitful if the feature selection reduces the learnable parameters. These algorithms often involve large search spaces and multiple iterations, so measuring runtime helps evaluate scalability, resource usage, and feasibility alongside accuracy; in this regard, the computational time optimized with SDGM, S-PSO and QBPSO is presented in Figure 11. It can be seen from Figure 11a and Figure 11b that the computational time lies in the range 150 to 300 s, 400 to 600 s and 600 to 800 s for ISIC2019 dataset on SDGM, S-PSO and QBPSO, respectively. Similarly, the mean computational time (mCT) for the HAM10000 dataset is found to be 153.782 s, 248.932 s and 353.279 s for SDGM, S-PSO and QBPSO, respectively.

### 4.2. Comparisons and Discussion of the Results

The results of the proposed framework are compared with state-of-the-art pre-trained models in the literature, various versions of the EffceinetNet-B0 and reposted results on SCLs classification.

#### 4.2.1. Comparison of the Proposed Model with Pre-Trained Models

The comparative analysis of different DL models highlights significant differences in terms of learnable parameters, network depth, and storage size is presented in Table 10. VGG-19, with 143.7 million parameters and 19 layers, is the heaviest model, requiring around 550 MB of storage, which reflects its computationally expensive nature. DarkNet-19 is much lighter, with only 20 million parameters across 19 layers and a compact size of 12 MB, making it more efficient for faster deployment. ResNet-50 and Inception-V3 strike a balance, with 25.6 million and 24.1 million parameters, respectively, and deeper structures (50 and 48 layers), resulting in moderate storage sizes of about 100 MB and 95 MB. In contrast, the modified EfficientNet-B0 (MENet-B0) optimized with swarm intelligence shows superior efficiency. The S-PSO optimized variant maintains 5.3 million parameters over 82 layers with just 29 MB in size, while the QBPSO optimized variant further reduces the parameter count to 4.9 million and storage size to 24.2 MB without sacrificing depth. This demonstrates the effectiveness of optimization in achieving lightweight yet powerful models, making MENet-B0 highly suitable for resource-constrained environments such as medical imaging applications like HAM10000 and ISIC2019. It is also worth mentioning that there was a very small difference in the performance measures of both the datasets.

#### 4.2.2. Comparison of the Proposed Model with State-of-the-Art Reported Results

A comprehensive comparison is made in the form of Table 11, Table 12 and Table 13 between the proposed architecture optimized with S-PSO and QBPSO algorithms and state-of-the-art reported results. The comparison of reported references highlights the performance of various approaches on the HAM10000, ISIC2019 and MSLD v2.0 skin lesion datasets in terms of average accuracy and FNR. For the HAM10000 dataset, earlier studies such as Khan et al. [28] and Ali et al. [22] achieved accuracies of 88.5% and 87.9% with corresponding FNRs of 11.5% and 12.1%. Subsequent works demonstrated gradual improvements, with Zafar et al. [23] reaching 92.7% accuracy, and Chanrvedi et al. [24] slightly improving to 93.0%. More recent contributions such as Venugopal [25], Sukanya et al. [29], and Tahir et al. [52] further increased accuracy to around 94%, reducing FNRs to nearly 5–6%. On the ISIC2019 dataset, reported results varied, with Pacheco et al. [26,30] achieving 90.1% in 2019 but dropping to 82.5% in 2020, while later works like Naeem et al. [31], Gilani et al. [53], and Zhou et al. [32] obtained more consistent performance ranging from 91% to 94% accuracy. In contrast to these results, the proposed MENet-B0 optimized with swarm-based algorithms achieved remarkable improvements. With S-PSO, MENet-B0 obtained 98.78% accuracy (1.22% FNR) on HAM10000 and 91.09% accuracy (8.91% FNR) on ISIC2019. The QBPSO variant further enhanced performance, reaching 99.62% accuracy with only 0.38% FNR on HAM10000 and 92.50% accuracy with 7.50% FNR on ISIC2019. These results clearly demonstrate the superiority of the optimized MENet-B0 models, significantly outperforming previously reported methods in terms of both accuracy and error reduction.

More explicitly, Table 13 shows that the proposed MENet-B0 models optimized with S-PSO and QBPSO outperform all compared SOTA methods, achieving the highest accuracy (96.39% and 98.30%) and the lowest FNRs (3.61% and 1.70%), clearly demonstrating their superior performance on the MSLD v-2.0 dataset.

#### 4.2.3. Comparison of Proposed Model with Other State-of-the-Art Reported Results EfficientNet Architectures

The comparative evaluation of different EfficientNet variants on the HAM10000 and ISIC2019 datasets shows a progressive improvement in accuracy with the introduction of optimized architectures, as shown in Table 14.

Standard EfficientNetV2 achieved accuracies of 93.78% on HAM10000 and 89.46% on ISIC2019, while EfficientNet-Lite and EfficientNet with NAS slightly improved results, reaching 94.89% and 95.23% on HAM10000, and around 89.9–90.0% on ISIC2019. Med-EfficientNet further enhanced performance, attaining 96.23% on HAM10000 and 90.71% on ISIC2019, highlighting the benefit of medical domain-specific adaptations. However, the proposed MENet-B0 models optimized with swarm intelligence techniques significantly outperformed these baselines. MENet-B0 with S-PSO achieved 98.78% accuracy on HAM10000 with a very low fval of 6.45 × 10^−10^ and a mCT of 248.932 s, and 91.09% accuracy with fval 3.89 × 10^−8^ and mCT 353.279 s on ISIC2019. The QBPSO optimized variant further improved results, achieving the highest accuracy of 99.62% on HAM10000 and 92.50% on ISIC2019, though with higher computational costs of 501.431 s and 752.421 s, respectively. This analysis demonstrates that MENet-B0 optimized with population-based algorithms not only surpasses existing EfficientNet variants in accuracy but also provides robust error minimization, albeit at the expense of increased computational time.

#### 4.2.4. Statistical Analysis Based on 100 Independent Runs Using Different Statistical Parameters

The statistical analysis in Table 15 summarizes the performance of the proposed MENet-B0 model optimized with S-PSO and QBPSO over 100 independent runs. The results indicate that the QBPSO consistently achieves lower objective function values (fval), with a minimum of 1.25 × 10^−10^ compared to 9.89 × 10^−10^ for S-PSO, reflecting its superior convergence capability. However, this improved accuracy comes at the cost of higher computational time, as QBPSO requires up to 400.23 s in the worst case, while S-PSO completes within 300.39 s. The mean performance (μ) also favors QBPSO, yielding 2.39 × 10^−10^ versus 6.45 × 10^−10^ for S-PSO, supported by smaller standard deviation values that highlight better stability. Kurtosis values for both algorithms remain close to 2, indicating moderately peaked distributions of results. Furthermore, the MaCC demonstrates that QBPSO (0.9580) provides more reliable classification accuracy than S-PSO (0.9230), confirming the effectiveness of QBPSO in balancing optimization precision with robustness, albeit with higher runtime overhead.

In order to have a deep analysis on the performance of proposed DL models like MENet-B0 optimized with S-PSO and MENet-B0 optimized with QBPSO, two statistical tests named analysis of variance (ANOVA) and *t*-test are applied on stored 100 independent runs data. The output of the one-way ANOVA is found to be 9.438 for the F statistic and *p*-value of 0.0153 that leads towards rejection of the null hypothesis. It is quite evident from Table 15 that there is a statistically significant difference between the methods. The results of the *t*-test are found to be −3.072 as t-statistics and a *p*-value of 0.0153 is observed that shows MENet-B0 optimized with QBPSO perform better than MENet-B0 optimized with the S-PSO technique. It is worth mentioning that with a confidence of 95% the MENet-B0 optimized with QBPSO performs better than that of the MENet-B0 optimized with S-PSO between 0.22 and 1.52 units.

The results can be reproduced by obtaining the open source, publicly available datasets, links to which are provided in the data availability statement of this paper. Moreover, the parameter values and settings for the proposed architecture MENet-B0 and QBPSO algorithm are provided in Table 3 and Table 4, respectively, in order to regenerate the exact proposed solution. The source code is part of a funded laboratory project, therefore, it cannot be publicly released; however, limited implementation details may be shared upon reasonable request.

#### 4.2.5. Limitations of Proposed Framework

Despite the various strengths of DL models for SCL classification, there are still limitations in the context of its reliable implementation in real clinical settings. Moreover, the images in benchmark datasets are typically captured under controlled dermoscopic conditions, whereas real-world clinical images vary greatly in quality, lighting, and device type, creating a domain shift that reduces diagnostic accuracy. Deep learning models also require large computational resources, making real-time deployment in hospitals or mobile applications challenging. Furthermore, these models function largely as black boxes, offering predictions without transparent reasoning, which reduces clinical trust and complicates regulatory approval. The presence of noisy or inconsistent labels in datasets, combined with the small size of many collections, further increases the risk of overfitting.

## 5. Conclusions

Based on the comprehensive scenario-based results calculated in the previous section, the following conclusion can be drawn:The proposed MENet-B0 architecture is validated on HAM10000 and ISIC2019 public datasets with mean accuracy of 99.62% and 92.50%, respectively, with an mCT of 501.431 and 752.421 s. Per class accuracy ranged from 99.45% (AKIC: 1989/2000) to 99.81% (MV: 6692/6705), with BCC and MEL also exceeding 99.5%, while misclassifications were extremely minimal and uniformly distributed, confirming the robustness and reliability of the model. Similarly for the ISIC 2019 dataset, the model achieved an overall accuracy of 92.50%, correctly classifying 37,811 cases and misclassifying 3064. Class-wise, VasL (95.67%), BaCC (95.53%), and DerF (94.95%) showed the best performance, while MN (90.66%) and BeKev (89.98%) were the weakest; moreover, Mel (93.38%) and ActK (92.17%) achieved strong mid-range accuracy. The overall accuracy of the proposed architecture for MSLD v2.0 is also found to be 98.30, outperforming the reported results.The fitness function fval value of HAM10000 lies in the range from 10^−2^ to 10^−4^, 10^−5^ to 10^−6^ and 10^−9^ to 10^−10^ for SDGM, S-PSO and QBPSO, respectively, to tune MENet-B0 architecture. Similarly, For the ISIC2019 dataset, the fval is observed to be in the range 10^−1^ to 10^−3^, 10^−4^ to 10^−6^ and 10^−7^ to 10^−9^ for SDGM, S-PSO and QBPSO, respectively, which shows the dominance of the QBPSO optimizer at the cost of a slightly high computational budget.The comparison highlights that earlier works on HAM10000 achieved accuracies around 88–94% with FNRs between 5 and 12%, while ISIC2019 results varied from 82% to 94%. In contrast, the proposed MENet-B0 optimized with S-PSO reached 98.78% accuracy (1.22% FNR) on HAM10000 and 91.09% on ISIC2019, whereas QBPSO achieved the highest results of 99.62% (0.38% FNR) and 92.50% (7.50% FNR), clearly outperforming prior methods.EfficientNet variants that are light weight models achieved up to 96.23% accuracy on HAM10000 and 90.71% on ISIC2019, while MENet-B0 with S-PSO reached 98.78% and 91.09%, respectively. The QBPSO variant further improved to 99.62% on HAM10000 and 92.50% on ISIC2019, though with higher computational times (501–752 s), showing a trade-off between accuracy and efficiency.QBPSO achieves superior optimization accuracy with lower objective function values and better stability than S-PSO, though it requires more computational time. Both methods show similar kurtosis values around 2, while QBPSO attains a higher MaCC (0.958 vs. 0.923), indicating more reliable classification despite higher runtime overhead.

In the future, we will implement the architecture on embedded hardware to use in the laboratories as testbed.

## Figures and Tables

**Figure 1 diagnostics-15-03245-f001:**
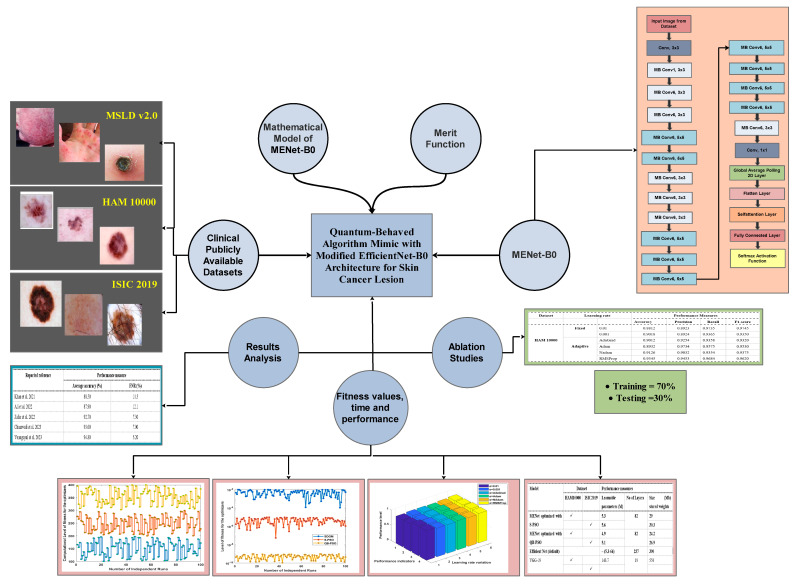
The overall proposed framework for skin cancer multi-class classification [22,23,24,25,28].

**Figure 2 diagnostics-15-03245-f002:**
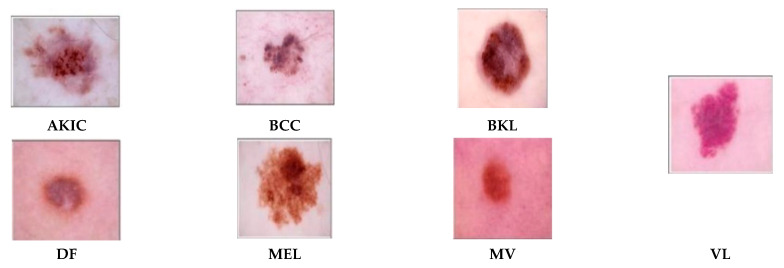
Skin lesion types extracted from the HAM10000 dataset.

**Figure 3 diagnostics-15-03245-f003:**
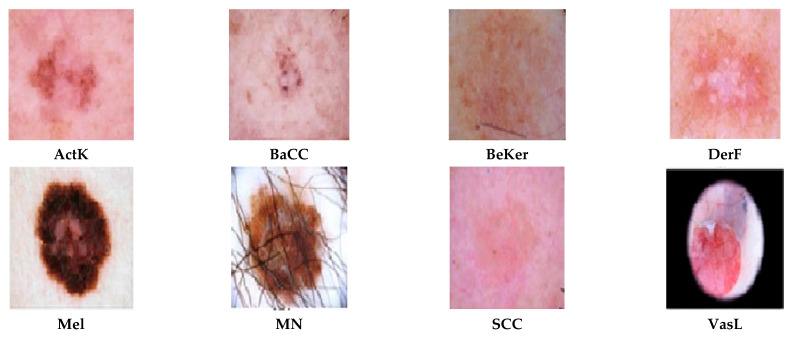
Skin lesion types extracted from the ISIC2019 dataset.

**Figure 4 diagnostics-15-03245-f004:**
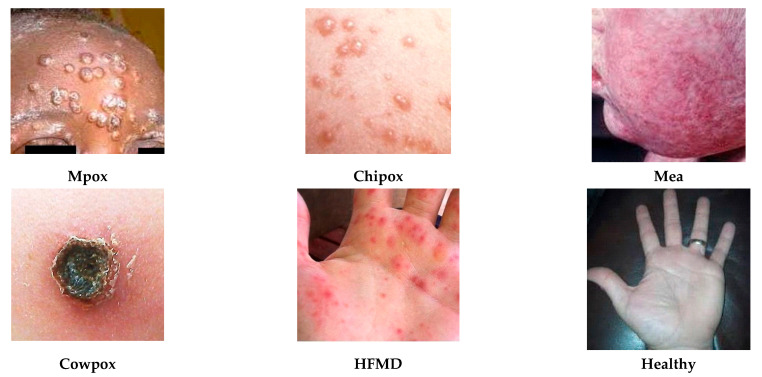
Few samples of SCL extracted from the MSLD v2.0 dataset.

**Figure 5 diagnostics-15-03245-f005:**
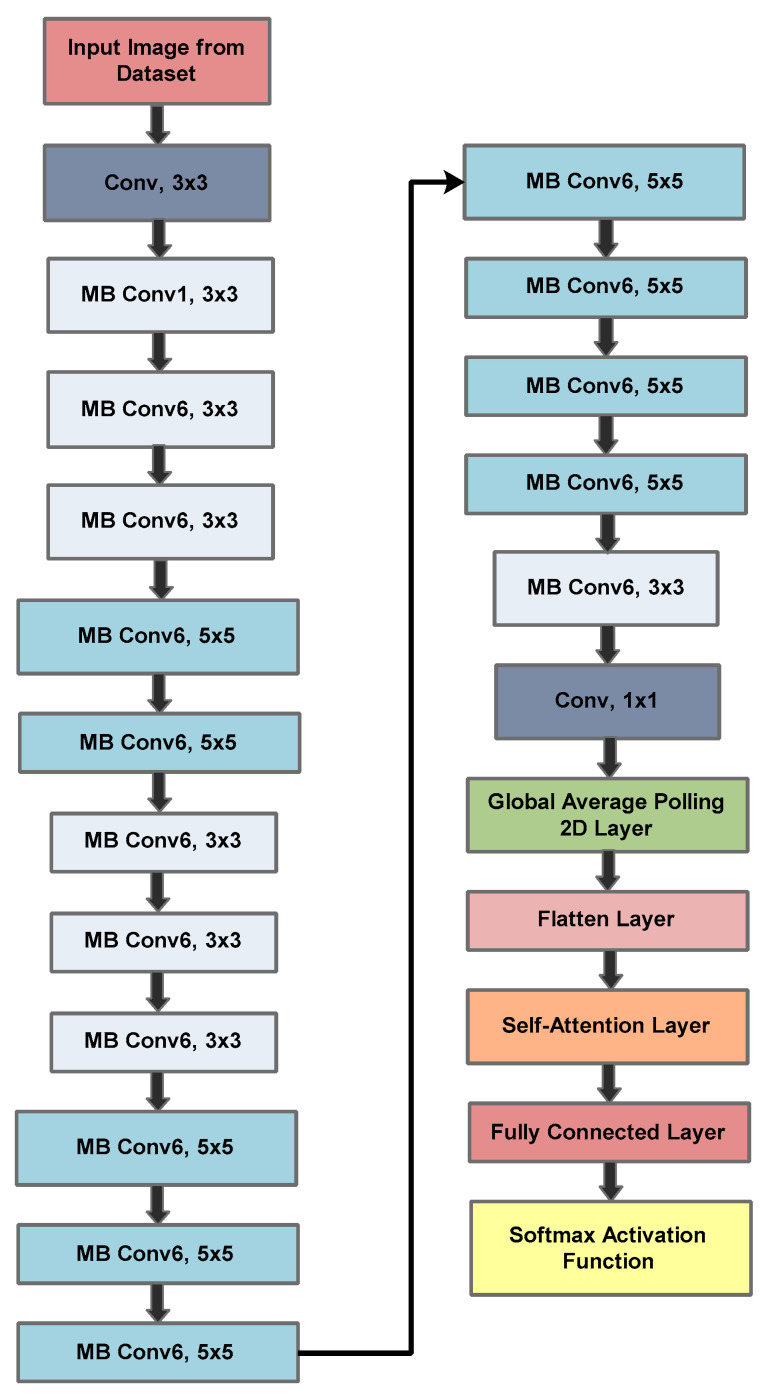
The proposed MENet-B0 architecture.

**Figure 6 diagnostics-15-03245-f006:**
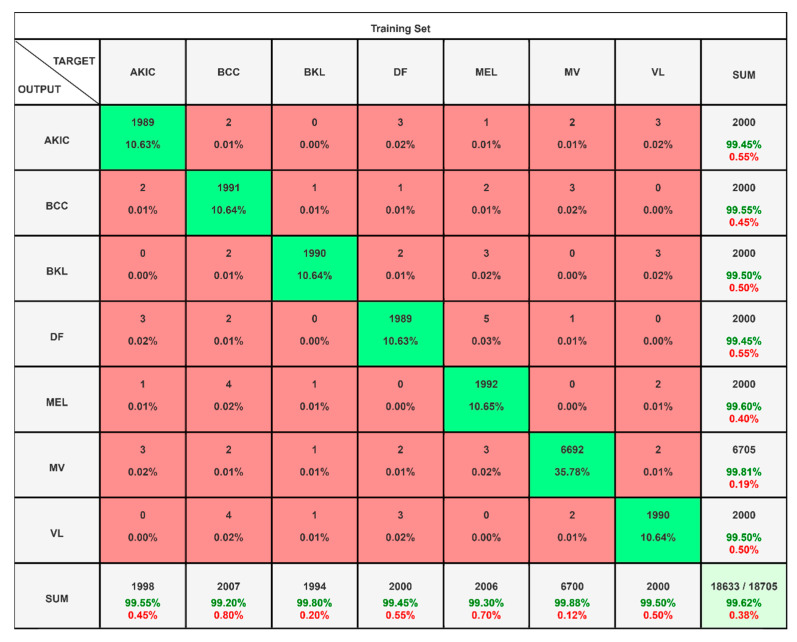
Confusion matrix highlighting exceptional accuracy and balanced predictions for HAM10000.

**Figure 7 diagnostics-15-03245-f007:**
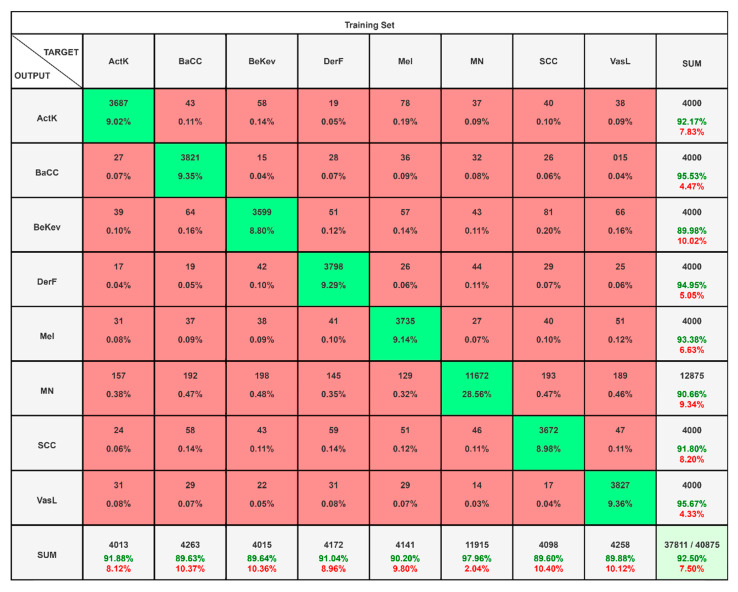
Confusion matrix highlighting exceptional accuracy and balanced predictions for ISIC2019.

**Figure 8 diagnostics-15-03245-f008:**
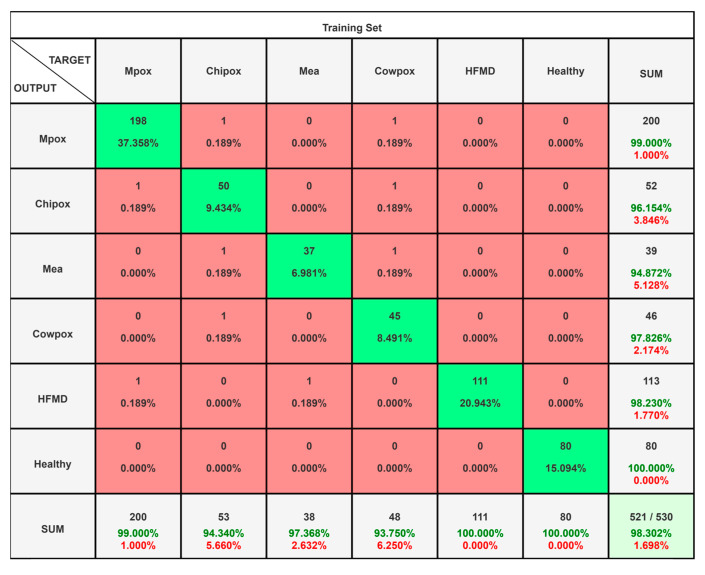
Confusion matrix representing the classification for MSLD v2.0 dataset.

**Figure 9 diagnostics-15-03245-f009:**
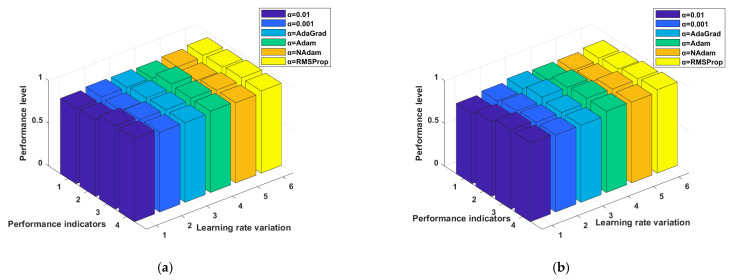
Comparative impact of learning rate strategies on proposed model accuracy. (**a**) Accuracy, precision, recall and F1 score achieved with different learning rates for the HAM10000 dataset, (**b**) Accuracy, precision, recall and F1 score achieved with different learning rates for the ISIC2019 dataset.

**Figure 10 diagnostics-15-03245-f010:**
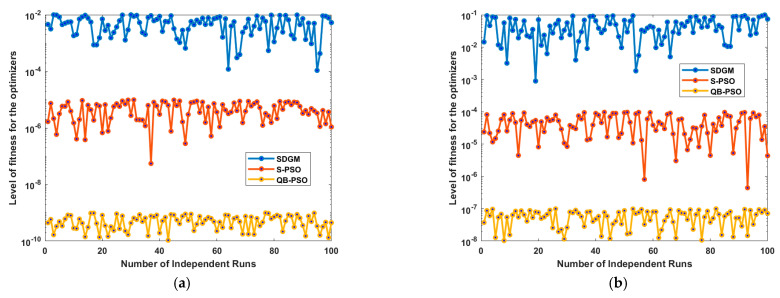
Level of the fitness values achieved by different optimizers. (**a**) Fitness values achieved for HAM10000; (**b**) Fitness values achieved for ISIC2019.

**Figure 11 diagnostics-15-03245-f011:**
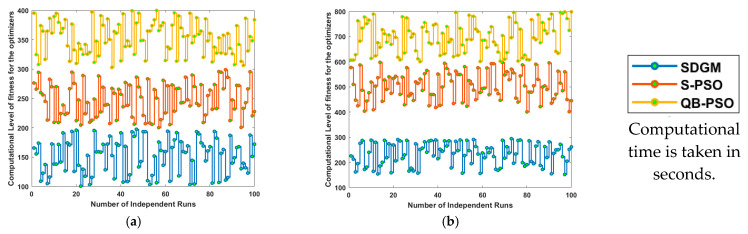
Computational time analysis optimized with various optimizers. (**a**) Computational time for the HAM10000 dataset on 100 runs, (**b**) Computational time for ISIC2019 dataset on 100 runs.

**Table 2 diagnostics-15-03245-t002:** Distribution of dataset and description for skin cancer lesion datasets.

Dataset	Classes	Training Data	Testing Data	Total Data	Size	Bit Depth
**HAM10000**	AKIC	1400	600	2000	224 × 224	24 Bits
	BCC	1400	600	2000		
	BKL	1400	600	2000		
	DF	1400	600	2000		
	MV	4694	2011	6705		
	MEL	1400	600	2000		
	VL	1400	600	2000		
**ISIC2019**	ActK	2800	1200	4000	224 × 224	24 Bits
	BaCC	2800	1200	4000		
	BeKer	2800	1200	4000		
	DerF	2800	1200	4000		
	Mel	2800	1200	4000		
	MN	9013	3862	12,875		
	SCC	2800	1200	4000		
	VasL	2800	1200	4000		
**MSLD v2.0**	Mpox	198	86	284	224 × 224	24 Bits
	ChiPox	52	23	75		
	Mea	39	16	55		
	Cowpox	46	20	66		
	HFMD	113	48	161		
	Healthy	80	34	114		

**Table 3 diagnostics-15-03245-t003:** Optimized hyperparameters for MENet-B0.

Hyperparameters Details	Optimized Values/Setting
Dropout Rate	0.2
Weight decay (L2 Regularization)	0.00001
K-fold cross-validation	10
Learning rate	0.0001
Mini batch size	16
Activation function	Swish
Optimizer	S-PSO, QBPSO
No of Epochs	40
Others	Default

**Table 4 diagnostics-15-03245-t004:** Adaptive parameters values and setting of QBPSO.

QBPSO Algorithm Parameters	Values/Setting
No of particles	100
Max^iter^	1000
Attraction coefficient	0.85
Contraction extension coefficient	0.80
Normalized scaling	[−1, 1]
*f* ^val^	≤10^−12^
Function tolerance	10^−10^
Others	Default

**Table 5 diagnostics-15-03245-t005:** Effectiveness of proposed framework on HAM10000 dataset.

Class	Performance Measures					
	Precision	Recall	FNR	F1 Score	Specificity	FPR
**AKIC**	0.9955	0.9945	0.0055	0.9950	0.9995	0.0005
**BCC**	0.9920	0.9955	0.0045	0.9938	0.9990	0.0010
**BKL**	0.9980	0.9950	0.0050	0.9965	0.9998	0.0002
**DF**	0.9945	0.9945	0.0055	0.9945	0.9993	0.0007
**MEL**	0.9930	0.9960	0.0040	0.9945	0.9992	0.0008
**MV**	0.9988	0.9981	0.0019	0.9984	0.9993	0.0007
**VL**	0.9950	0.9950	0.0050	0.9950	0.9994	0.0006

**Table 6 diagnostics-15-03245-t006:** Effectiveness of proposed framework on ISIC2019 dataset.

Class	Performance Measures					
	Precision	Recall	FNR	F1-Score	Specificity	FPR
**ActK**	0.9188	0.9217	0.0783	0.9203	0.9912	0.0088
**BaCC**	0.8963	0.9553	0.0447	0.9248	0.9880	0.0120
**BeKev**	0.9864	0.8998	0.1002	0.8881	0.9887	0.0113
**DerF**	0.9104	0.9496	0.0505	0.9285	0.9899	0.0101
**Mel**	0.9020	0.9337	0.0683	0.9176	0.9890	0.0110
**MN**	0.9796	0.9066	0.0934	0.9417	0.9913	0.0087
**SCC**	0.8960	0.9180	0.0820	0.9009	0.9880	0.0120
**VasL**	0.8988	0.9567	0.0433	0.9269	0.9883	0.0117

**Table 7 diagnostics-15-03245-t007:** Effectiveness of proposed framework on MSLD v2.0 dataset.

Class	Performance Measures					
	Precision	Recall	FNR	F1-Score	Specificity	FPR
**Mpox**	0.9900	0.9900	0.0100	0.9900	0.9939	0.0061
**Chipox**	0.9434	0.9615	0.0385	0.9524	0.9937	0.0063
**Mea**	0.9737	0.9487	0.0513	0.9610	0.9980	0.0020
**Cowpox**	0.9375	0.9783	0.0217	0.9574	0.9938	0.0062
**HFMD**	1.0000	0.9823	0.0177	0.9911	1.0000	0.0000
**Healthy**	1.0000	1.0000	0.0000	1.0000	1.0000	0.0000

**Table 8 diagnostics-15-03245-t008:** Impact of the learning rate on the effectiveness of the proposed framework.

Dataset	Learning Rate		Performance Measures			
			Accuracy	Precision	Recall	F1-Score
**HAM10000**	**Fixed**	0.01	0.8812	0.8923	0.9735	0.9745
		0.001	0.9018	0.8924	0.9365	0.9350
	**Adaptive**	AdaGrad	0.9012	0.9254	0.9358	0.9320
		Adam	0.8932	0.9734	0.9575	0.9530
		Nadam	0.9126	0.9032	0.9354	0.9375
		RMSProp	0.9345	0.9453	0.9684	0.9620
**ISIC2019**	**Fixed**	0.01	0.8235	0.8734	0.9274	0.9240
		0.001	0.8534	0.8855	0.9024	0.9024
	**Adaptive**	AdaGrad	0.8835	0.9353	0.90193	0.9023
		Adam	0.8734	0.9346	0.9565	0.9525
		Nadam	0.8734	0.8932	0.9492	0.9448
		RMSProp	0.9023	0.9219	0.9550	0.9750

**Table 9 diagnostics-15-03245-t009:** Evaluation of EfficientNet Architectures Versus MENet-B0 for SCLs classification.

Model Version	HAM10000 Dataset		ISIC2019	
	Average Accuracy	*f* ^val^	Average Accuracy	*f* ^val^
EfficientNet-B0	97.37	5.3756 × 10^−9^	82.35	5.7367 × 10^−7^
EfficientNet-B1	97.34	7.4636 × 10^−9^	83.09	7.3457 × 10^−7^
EfficientNet-B2	97.39	8.6732 × 10^−9^	83.89	8.4835 × 10^−7^
EfficientNet-B3	97.65	4.4756 × 10^−9^	85.23	4.1235 × 10^−8^
EfficientNet-B4	97.39	1.4857 × 10^−9^	87.67	1.1282 × 10^−8^
EfficientNet-B5	97.65	4.6732 × 10^−9^	87.92	4.3465 × 10^−8^
EfficientNet-B6	97.97	5.9675 × 10^−10^	88.26	5.7367 × 10^−8^
EfficientNet-B7	98.12	3.4966 × 10^−10^	88.99	3.4753 × 10^−8^
**MENet-B0 optimized with S-PSO**	**98.78**	**6.4593 × 10^−10^**	**91.09**	**3.8933 × 10^−8^**
**MENet-B0 optimized with QBPSO**	**99.62**	**2.3912 × 10^−10^**	**92.50**	**1.7921 × 10^−8^**

**Table 10 diagnostics-15-03245-t010:** Model parameters, depth, and memory footprint of pre-trained models and optimized MENet-B0.

Model	Dataset		Performance Measures		
	HAM10000	ISIC2019	Learnable Parameters (M)	No. of Layers	Size (Mb) Stored Weights
**MENet optimized with S-PSO**	**✓**		5.3	82	29
		**✓**	5.6		30.3
**MENet optimized with QBPSO**	**✓**		4.9	82	24.2
		**✓**	5.1		26.9
**Efficient Net (default)**			~(5.3–66)	237	390
VGG-19	**✓**		143.7	19	550
		**✓**			
DarkNet-19	**✓**		20	19	12
		**✓**			
ResNet-50	**✓**		25.6	50	100
		**✓**			
Inception-V3	**✓**		24.1	48	95
		**✓**			

**Table 11 diagnostics-15-03245-t011:** Comparative study of state-of-the-art (SOTA) techniques with the proposed framework using HAM-10000.

Reported Reference	Performance Measure	
	Average Accuracy (%)	FNR (%)
Khan et al. [28], 2021	88.50	11.5
Ali et al. [22], 2022	87.90	12.1
Zafar et al. [23], 2022	92.70	7.30
Chanrvedi et al. [24], 2023	93.00	7.00
Venugopal, V. [25], 2023	94.80	5.20
Sukanya et al. [29], 2023	94.00	6.00
Tahir et al. [52], 2023	94.17	5.83
**MENet-B0 optimized with S-PSO**	**98.78**	**1.22**
**MENet-B0 optimized with QBPSO**	**99.62**	**0.38**

**Table 12 diagnostics-15-03245-t012:** Comparative study of the proposed framework with SOTA using ISIC-2019.

Reported Reference	Performance Measure	
	Average Accuracy (%)	FNR (%)
Pacheco et al. [30], 2019	90.1	9.90
Pacheco et al. [26], 2020	82.5	17.5
Naeem et al. [31], 2022	92.91	7.09
Aldhyani et al. [27], 2022	89.5	10.5
Gilani et al. [53], 2023	91.81	8.19
**MENet-B0 optimized with S-PSO**	**91.09**	**8.91**
**MENet-B0 optimized with QBPSO**	**92.50**	**7.50**

**Table 14 diagnostics-15-03245-t014:** Comparative analysis of different lightweight EfficientNets with the proposed framework.

Models	Performance Measure					
	Accuracy (%)		*f* ^val^		mCT (s)	
	HAM10000	ISIC2019	HAM10000	ISIC2019	HAM10000	ISIC2019
EfficientNetV2 [64]	93.78	89.46	-	-	-	-
EfficientNet-Lite [65]	94.89	89.93	-	-	-	-
EfficientNet hybrid with NAS [66]	95.23	90.71	-	-	-	-
Med-EfficientNet [67]	96.23	90.01	-	-	-	-
**MENet-B0 optimized with S-PSO**	**98.78**	**91.09**	**6.45 × 10^−10^**	**3.89 × 10^−8^**	**248.932**	**353.279**
**MENet-B0 optimized with QBPSO**	**99.62**	**92.50**	**2.39 × 10^−10^**	**1.79 × 10^−8^**	**501.431**	**752.421**

**Table 15 diagnostics-15-03245-t015:** Statistical analysis on 100 independent runs for the proposed model.

Statistical Parameters	Optimization Algorithm		Performance Measure	
	S-PSO	QBPSO	*f* ^val^	Computational Time (s)
**MIN**	**✓**		9.89 × 10^−10^	200.37
		**✓**	1.25 × 10^−10^	300.69
**MAX**	**✓**		9.96 × 10^−5^	300.39
		**✓**	9.89 × 10^−8^	400.23
** *µ* **	**✓**		6.45 × 10^−10^	250.46
		**✓**	2.39 × 10^−10^	344.99
**Std**	**✓**		2.89 × 10^−5^	30.2949
		**✓**	2.47 × 10^−8^	30.1790
**Kur**	**✓**		1.7578	1.7425
		**✓**	1.8873	1.6496
**MaCC**	**✓**		**0.923**	-
		**✓**	**0.958**	-

**Table 13 diagnostics-15-03245-t013:** Comparative study of proposed framework with SOTA using MSLD v-2.0.

Reported Reference	DL Model Used	Performance Measure	
		Average Accuracy (%)	FNR (%)
Waqar, M. et al. [49], 2025	ConvexNeXt Models	94.00	6.00
Elhadidy, M.S. et al. [50], 2025	Vision transformer	93.10	6.90
	Dino	90.40	9.60
	Swin Transformer	93.71	6.29
Alghoraibi, H. et al. [51], 2025	ResNetViT	92.00	8.00
	ViT Hybrid Model	92.19	7.81
**Proposed**	**MENet-B0 optimized with S-PSO**	**96.39**	**3.61**
	**MENet-B0 optimized with QBPSO**	**98.30**	**1.70**

## Data Availability

The data was accessed on 14 May 2024 that support the findings of this study are openly available at: https://www.kaggle.com/datasets/nour12347653/skin-disease-detection-dataset-ham10000-isic; https://www.kaggle.com/datasets/joydippaul/mpox-skin-lesion-dataset-version-20-msld-v20.

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
