# Peer review of "Modified EfficientNet-B0 Architecture Optimized with Quantum-Behaved Algorithm for Skin Cancer Lesion Assessment"

_diagnostics, 2025, doi:10.3390/diagnostics15243245_

Round 1

Reviewer 1 Report

Comments and Suggestions for Authors

1- The fact that the datasets used in the study are outdated and that it is a heavily researched topic is also the study's weakest point.

2- In my research, I observed that there are more recent datasets in this fields such as “Mpox Skin Lesion Dataset Version 2.0 (MSLD v2.0)”. There are also research articles published on this topic in the Diagnostics journal. I recommend expanding your work with a more up-to-date dataset.

3- I am unsure about the placement of your “Nomenclature” section. I recommend reviewing the MDPI writing format again.

4- The studies in Table 1 are from 2023, which is the most recent year. You need to add more recent studies to your manuscript.

5- The mathematical formulas in Fig. 1 are incomprehensible in their current form. The authors should consider moving the formulas into the text. You should add explanations such as symbol descriptions.

6- It would be more useful if you could also show the validation subset in Table 2.

7- In “Table 9: Model parameters, depth, and memory footprint of pre-trained models and optimized MENet-B0,” you should also provide the number of parameters for the EfficientNet architecture in its default state so that the comparison is more accurate.

8- Tables 10 and 11 are very confusing. It may be more appropriate to share the data sets as separate tables. As it stands, we have to follow them line by line.

9- You may also consider performing a statistical confidence test. It may be useful to analyze model confidence by adding a method such as ANNOVA - TUKEY HSD.

10- A discussion regarding the repeatability and reproducibility of the study should be added to the content. The use of the data set and the visibility of the codes can be emphasized at this stage.

11- If possible, a time analysis should be added to the study, and the training time of the architecture should be provided within the study. The time analysis differences between the architectures should be compared.

Author Response

Comments 1: The fact that the datasets used in the study are outdated and that it is a heavily researched topic is also the study's weakest point.

Response 1: Thank you for pointing such an important aspect in this regard the latest dataset with updated version called Mpox Skin Lesion Dataset version 2.0 is included in the study to improve the technical strength of the article. (Page 5 last paragraph, Table-1 is updated on Page 6).

Comments 2: In my research, I observed that there are more recent datasets in this fields such as “Mpox Skin Lesion Dataset Version 2.0 (MSLD v2.0)”. There are also research articles published on this topic in the Diagnostics journal. I recommend expanding your work with a more up-to-date dataset.

Response 2: We are thankful to the reviewer for this in-depth comment, in this regards the study has been expanded by considering the updated dataset called “Mpox Skin Lesion Dataset Version 2.0 (MSLD v2.0)”. Moreover, the latest contributions of 2025 have also been included in the study as highlighted by the worthy reviewer (Section 2- Literature Survey and State of Art, Page 5 last paragraph, Table-1 is updated on Page 6):

 Waqar, Muhammad, Zeshan Aslam Khan, Shanzey Tariq Khawaja, Naveed Ishtiaq Chaudhary, Saadia Khan, Khalid Mehmood Cheema, Muhammad Farhan Khan, Syed Sohail Ahmed, and Muhammad Asif Zahoor Raja. "Explainable clinical diagnosis through unexploited yet optimized fine-tuned ConvNeXt models for accurate monkeypox disease classification." SLAS technology (2025): 100336.

Vuran, Seyfettin, Murat Ucan, Mehmet Akin, and Mehmet Kaya. "Multi-classification of skin lesion images including Mpox disease using transformer-based deep learning architectures." Diagnostics 15, no. 3 (2025): 374.

Alghoraibi, Huda, Nuha Alqurashi, Sarah Alotaibi, Renad Alkhudaydi, Bdoor Aldajani, Joud Batawil, Lubna Alqurashi, Azza Althagafi, and Maha A. Thafar. "Deep Learning-Based Mpox Skin Lesion Detection and Real-Time Monitoring in a Smart Healthcare System." Diagnostics 15, no. 19 (2025): 2505.

These references are also included in the reference section on Page 30.

Comments 3: I am unsure about the placement of your “Nomenclature” section. I recommend reviewing the MDPI writing format again.

Response 3: We have been gone through the MDPI “Layout Style Guide” and found that they have put the position of the nomenclature on the discretion of the author (s) to ensure the clarity and readability of the manuscript. However, if worthy reviewer wants then we can put it at his suggested position.

Comments 4: The studies in Table 1 are from 2023, which is the most recent year. You need to add more recent studies to your manuscript.

Response 4: In this regard three recent and relevant studies of 2025 have been included in the manuscript. The authors are really thankful to the reviewer in enhancing the quality of the article. The compliance of this point can be found at following positions in the manuscript. (Section 2- Literature Survey and State of Art, Page 5 last paragraph, Table-1 is updated on Page 6). The most recent studies are also referenced on Page 30.

Comments 5: The mathematical formulas in Fig. 1 are incomprehensible in their current form. The authors should consider moving the formulas into the text. You should add explanations such as symbol descriptions.

Response 5: The mathematical formulas in Fig. 1 are removed from the Fig. 1 and are explained on Page 10-11 as Eq. (1) to (4), Eq. (7) and (8) as per guidance that seems to be sound.   

Comments 6: It would be more useful if you could also show the validation subset in Table 2.

Response 6: In this study the validation is performed through the testing data that is 30% of the total images and are provided in Table 2 of each dataset and for each sub-class.

Comments 7: In “Table 9: Model parameters, depth, and memory footprint of pre-trained models and optimized MENet-B0,” you should also provide the number of parameters for the EfficientNet architecture in its default state so that the comparison is more accurate.

Response 7: Complied as guided in order to develop more clarity and comparison. The table 9 is updated now as Table 10 in the revised manuscript by adding the default values of (Model parameters, depth, and memory footprint) for default setting of EfficientNet architecture. (Page 23, Table 10)  

Comments 8: Tables 10 and 11 are very confusing. It may be more appropriate to share the data sets as separate tables. As it stands, we have to follow them line by line.

Response 8: In order to make the clarity the Table 10 is split into three separate tables (Now Table 11, 12 and 13 in the revised manuscript) keeping in view the three datasets. (Page 24)

The Table 11 is remodeled now as Table 14 (revised manuscript) so that it can followed line by line on Page 25.  

Comments 9: You may also consider performing a statistical confidence test. It may be useful to analyze model confidence by adding a method such as ANOVA - TUKEY HSD.

Response 9: The authors are thankful for this suggestion and is incorporated in the manuscript by applying the ANOVA and t-test to see the performance of the DL proposed models along with its confidence interval on Page 26, 1st paragraph right after the Table 15. 

Comments 10: A discussion regarding the repeatability and reproducibility of the study should be added to the content. The use of the data set and the visibility of the codes can be emphasized at this stage.

Response 10: A complete paragraph before the conclusions section has been added on Page 26 to highlight the reproducibility of the study, use of dataset and visibility of code. (Second last paragraph before the conclusion section on Page 26)

Comments 11: If possible, a time analysis should be added to the study, and the training time of the architecture should be provided within the study. The time analysis differences between the architectures should be compared.

Response 11: The computational time analysis is provided in the last column of Table 15 on Page 26 for 100 independent runs, moreover the mean computation time is also provided keeping in view the stochastic nature of the deep learning techniques.

4. Response to Comments on the Quality of English Language

Point 1: The English is fine and does not require any improvement.

Response 1: We are thankful to the reviewer for this encouragement. However, the complete manuscript is checked for typo and grammatical correction again.

5. Additional clarifications

We are Thankful for the editor (s) and reviewers for their time and valuable comments to improve the quality of the manuscript.

Reviewer 2 Report

Comments and Suggestions for Authors

well done for this great and well organised work on the modified  EfficientNet-B0 Architecture used for models for skin cancer diagnosis

comments:

1) report more on the modifivation type such as  inverted bottleneck convolution with squeeze and excitation terminology in the abstract

2) create a figure/ blots  for the skin cancer diagnosis accuracy

3)Does the model outperform other CNN architectures on skin-cancer detection- make it more clear

4)report a limitation paragraph and create a table comparing the datasets

5) the most of the manuscript is detailed with excellent tables, analytical desciptions of processess, tables with important study presentations .... very very good impression and excellent manuscript build up  

Author Response

Comments 1: Report more on the modification type such as inverted bottleneck convolution with squeeze and excitation terminology in the abstract.

Response 1: Complied as required on Page 1 by providing more detail about inverted bottleneck convolution with squeeze and excitation.  

Comments 2: Create a figure/ plot for the skin cancer diagnosis accuracy.

Response 2: Agree for the better representation of the results. In this regard Fig. 9 presents the level of accuracy achieved for HAM10000 and ISIC2019 dataset at different learning techniques and rates in skin cancer disease classification, respectively. (Page 20).

Comments 3: Does the model outperform other CNN architectures on skin-cancer detection- make it more clear.

Response 3: The proposed architecture outperforms as compared with different reported results in Table 9, 10, 11, 12, 13 and 14, respectively on Page 21 to Page 25.

Comments 4: Report a limitation paragraph and create a table comparing the datasets

Response 4: The limitation paragraph has been added on Page 26 in the subheading of “Limitations of proposed framework” before the conclusion sections. Moreover, the comparison table for different dataset is addressed in Table 11,12 and 13, respectively.

Comments 5: The most of the manuscript is detailed with excellent tables, analytical descriptions of processes, tables with important study presentations .... very very good impression and excellent manuscript build up.   

Response 5: The authors once again thankful for such an encouraging comment of unknown reviewer.

4. Response to Comments on the Quality of English Language

Point 1: The English is fine and does not require any improvement.

Response 1: We are thankful to the reviewer for this encouragement. However, the complete manuscript is checked for typo and grammatical correction again.

5. Additional clarifications

We are Thankful for the editor (s) and reviewers for their time and valuable comments to improve the quality of the manuscript.

Round 2

Reviewer 1 Report

Comments and Suggestions for Authors

I would like to thank the authors for carefully answering all the questions. As a reviewer, I am pleased to have been able to help the article reach a higher level. Thank you.